# OpenReview forum: "OCN: Learning Object-centric Representations for Unsupervised Multi-object Segmentation"
_ICLR.cc/2025/Conference — Submitted to ICLR 2025_

### Official Review · Reviewer_gBo7 · 2024-10-21

**Soundness:** 2
**Presentation:** 3
**Contribution:** 2
**Rating:** 6
**Confidence:** 5

**Summary:**

This paper introduces an unsupervised object discovery framework for real-world images, called OCN (Object-Centric Representations via the Objectness Network), which operates without human annotations. To achieve this, this paper proposes Objectness Network, trained on ImageNet using masks generated by CuVLER [1], and it produces three outputs: i) an Object Existence Score, which determines if an object is present in the image; ii) an Object Center Field, which estimates pixel locations relative to the object center; and iii) an Object Boundary Field, which estimates pixel distance to the object’s boundary. Then, it adopts a Multi-Object Reasoning Module, where bounding boxes are iteratively updated based on the Objectness Network’s outputs. OCN achieves state-of-the-art performance on the object discovery task across several datasets, including COCO and COCO*, an extended version with additional object annotations.


---
[1] Shahaf Arica et al., CuVLER: Enhanced Unsupervised Object Discoveries through Exhaustive Self-Supervised Transformers, CVPR 2024

**Strengths:**

* This paper revisits the important concept of object discovery by decomposing objectness into three object-centric representations, followed by a network-free multi-object reasoning module.
* The use of boundary distance gradients for extending and shrinking bounding boxes is particularly effective, as it is parameter-free and potentially faster.
* The model successfully captures multiple objects of varying scales in the challenging COCO dataset.
* The paper is well-written, easy to follow, and supported by clear explanatory figures.

**Weaknesses:**

* The model is more similar to MaskCut [1] supervised rather than unsupervised, as the objectness network is trained using masks from MaskCut. Therefore, the objectness network should be compared to CutLER [1] or CuVLER [2], but not to the pseudo-labeling mechanisms of them, ie MaskCut or VoteCut, as shown in Table 1. Since OCN includes an extra training step on ImageNet with pseudo-labels, whereas MaskCut and VoteCut do not involve training, this comparison is unfair. If we assume $g$ is the MaskCut operation and $I$ represents the data, Table 1 compares $g(I)$ to $f(g(I))$, where $f$ represents the OCN training.
* Similarly, Table 3 incorporates an additional level of training, giving OCN an advantage. The OCN results in this case are $p(f(g(I)))$, while the others are only $p(g(I))$, where $p$ is the detector training.
* The use of anchors and the existence network closely resembles the Region Proposal Network (RPN) in Faster R-CNN [3]. In this sense, the objectness network can be seen as a modified version of RPN (with additional outputs), which diminishes the novelty of the proposed network.
* The Objectness Network, particularly the center field module, is trained on images with single objects. However, it is used to detect multiple objects in an image, which is not represented during training. Is this possible due to the random cropping augmentation? The authors should provide more insights into this.

**Minor Comments**
* In the appendix, it states “MaskCut proposed in CuVLER,” but MaskCut was actually proposed in CutLER [1].
* The related work section on Object-Centric Learning with Pretrained Features is missing relevant references, such as SOLV [4] and VideoSAUR [5].
* There is no step #4 mentioned in the text, though it is referred to in line 306.
* Typos in Table 5: row 2 (“exsitence”) and row 6 (“filed”).


---
[1] Xudong Wang et al., Cut and Learn for Unsupervised Object Detection and Instance Segmentation, CVPR 2023

[2] Shahaf Arica et al., CuVLER: Enhanced Unsupervised Object Discoveries through Exhaustive Self-Supervised Transformers, CVPR 2024

[3] Shaoqing Ren et al., Faster R-CNN: Towards Real-Time Object Detection with Region Proposal Networks, arXiv

[4] Görkay Aydemir et al., Self-supervised object-centric learning for videos, NeurIPS 2023

[5] Andrii Zadaianchuk et al., Object-Centric Learning for Real-World Videos by Predicting Temporal Feature Similarities, NeurIPS 2023

**Questions:**

* Why didn’t the authors utilize DINOv2 [1], which has been shown to exhibit greater object awareness, instead of DINO [2]?
* On average, how many iterations does it take to form the final bounding box?
---
[1] Maxime Oquab et al., DINOv2: Learning Robust Visual Features without Supervision, arXiv

[2] Mathilde Caron et al., Emerging Properties in Self-Supervised Vision Transformers, ICCV 2021

---

> ### Author Response · Authors · 2024-11-24
> **Responses to Reviewer gBo7 (Part 1)**
>
> We appreciate the reviewer’s thoughtful comments and address the main concerns below.
>
> ***Q1: Weakness 1) The model is more similar to MaskCut [1] supervised rather than unsupervised, as the objectness network is trained using masks from MaskCut. Therefore, the objectness network should be compared to CutLER [1] or CuVLER [2], but not to the pseudo-labeling mechanisms of them, ie MaskCut or VoteCut, as shown in Table 1. Since OCN includes an extra training step on ImageNet with pseudo-labels, whereas MaskCut and VoteCut do not involve training, this comparison is unfair. If we assume $g$ is the MaskCut operation and $I$ represents the data, Table 1 compares $g(I)$ to $f(g(I))$, where $f$ represents the OCN training.***
>
> **A1**: We appreciate this thought-provoking comment. Unlike fully-supervised methods, the fairness of unsupervised experiment settings is primarily determined by the amount of unlabeled data and external information used for each method. From this point of view, in our experiments of Direct Object Discovery in Section 4.1, fundamentally, all baselines and our OCN$_{disc}$ just rely on the self-supervised pretrained features, without needing any human labels or training additional multi-object detectors.
>
> Following the reviewer's comment, if we regard the whole unlabeled ImageNet dataset as $\mathbb{I}$, a test COCO image as $I$, MaskCut or VoteCut operation as $g$, then the baseline method is:  $g(dino(\mathbb{I}, I))$. By contrast, for our method OCN$_{disc}$, our objectness network can be just regarded as an improved version of $g(dino(\mathbb{I}))$, denoted as $g'(dino(\mathbb{I}))$, because we never leverage any additional information beyond $g(dino(\mathbb{I}))$. The key difference is that: $g(dino(\mathbb{I}))$ relies on classic graph operations, while our $g'(dino(\mathbb{I}))$ relies on neural operations.
> Therefore, to test on a COCO image $I$, our full method is: $g'(dino(\mathbb{I}), I)$. This means that the experiment setting for direct object discovery in Section 4.1 is particularly fair. Actually, this setting follows the de facto standard in prior works CutLER and CuVLER in the field of unsupervised multi-object segmentation.
>
> For comparison with CutLER and CuVLER, all experiments are conducted in Section 4.2, where all methods are trained with additional detectors for a fair comparison.
>
> ***Q2: Weakness 2) Similarly, Table 3 incorporates an additional level of training, giving OCN an advantage. The OCN results in this case are $p(f(g(I))$, while the others are only $p(g(I)$, where $p$ is the detector training.***
>
> *A2*: For Table 3 in our original paper (now Table 2 in the revised paper),
> the baseline CutLER actually trains 3 detectors one by one in order to discover more objects and get the final scores. The baseline CuVLER trains 2 detectors one by one to obtain the final good scores. These training details can be found in their papers. By comparison, our OCN only trains one detector to obtain the final scores.
>
> Nevertheless, to respect the prior arts, we still argue the experiment setting is generally fair because there are no external supervision signals introduced during training detectors.
>
> ***Q3: Weakness 3) The use of anchors and the existence network closely resembles the Region Proposal Network (RPN) in Faster R-CNN [3]. In this sense, the objectness network can be seen as a modified version of RPN (with additional outputs), which diminishes the novelty of the proposed network.***
>
> **A3**: For clarification, our multi-object reasoning module does not rely on any specific strategy for initial proposal generation, as long as there is a sufficient number of proposals with diverse locations and sizes on the target image. And the proposal initialization is not our core novelty as well.
>
> To validate this, we conduct another experiment where we randomly and uniformly generate the same number of proposals as RPN. These proposals have various sizes and locations scattering on images. The following Table 1 shows the results of direct object discovery on COCO\* validation set. We can see that these two initialization strategies give almost the same level of performance.
>
> In the revised paper, to avoid confusion, we have rephrased the descriptions in Section 3.3.
>
> *Table 1: The comparison of two different ways for proposal initialization.*
> |  | $\mathrm{AP}_{50}^{\text {box}}$ | $\mathrm{AP}_{75}^{\text {box}}$ | $\mathrm{AP}^{\text {box}}$ | $\mathrm{AR}_{100}^{\text {box}}$ | $\mathrm{AP}_{50}^{\text {mask}}$ | $\mathrm{AP}_{75}^{\text {mask}}$ | $\mathrm{AP}^{\text {mask}}$ | $\mathrm{AR}_{100}^{\text {mask}}$ |
> |---|---|---|---|---|---|---|---|---|
> | RPN initialization | 19.1 | 9.0 | 10.1 | 19.6 | 17.8 | 8.7 | 9.5 | 18.9 |
> | Random initialization | 19.0 | 9.0 | 10.1 | 19.4 | 17.8 | 8.7 | 9.4 | 18.7 |

---

> ### Author Response · Authors · 2024-11-24
> **Responses to Reviewer gBo7 (Part 2)**
>
> ***Q4: Weakness 4) The Objectness Network, particularly the center field module, ... Is this possible due to the random cropping augmentation? The authors should provide more insights into this.***
>
> **A4**: Thanks for this insightful comment. To evaluate the effectiveness of random cropping augmentation for learning object-centric representations, we conduct an additional ablation study. In particular, we omit random cropping during training our objectness network, while keeping all other settings the same. The following Table 2 shows the results of direct object discovery on COCO\* validation set. We can see that random cropping is indeed helpful for the objectness network to learn robust center and boundary fields. Primarily, this is because during the multi-object reasoning stage, many proposals just have partial or fragmented objects, but the random cropping augmentation inherently enables the objectness network to infer rather accurate center and boundary fields for those partial objects, thus driving the proposals to be updated correctly.
>
> In the revised paper, we have added these ablation results in Table 12 of Appendix A.10.
>
> *Table 2: Ablation results on COCO\* validation set for random cropping augmentation of the objectness network.*
> |  | $\mathrm{AP}_{50}^{\text {box}}$ | $\mathrm{AP}_{75}^{\text {box}}$ | $\mathrm{AP}^{\text {box}}$ | $\mathrm{AR}_{100}^{\text {box}}$ | $\mathrm{AP}_{50}^{\text {mask}}$ | $\mathrm{AP}_{75}^{\text {mask}}$ | $\mathrm{AP}^{\text {mask}}$ | $\mathrm{AR}_{100}^{\text {mask}}$ |
> |---|---|---|---|---|---|---|---|---|
> | OCN$_{disc}$ (with random cropping) | 19.1 | 9.0 | 10.1 | 19.6 | 17.8 | 8.7 | 9.5 | 18.9 |
> | OCN$_{disc}$ (w/o random cropping) | 15.7 | 7.5 | 8.2  | 18.1 | 15.6 | 6.6 | 7.9 | 17.4 |
>
> ***Q5: Minor Comments 1) In the appendix, it states “MaskCut proposed in CuVLER,” but MaskCut was actually proposed in CutLER [1]. 2) The related work section on Object-Centric Learning with Pretrained Features is missing relevant references, such as SOLV [4] and VideoSAUR [5]. 3) There is no step \#4 mentioned in the text, though it is referred to in line 306. 4) Typos in Table 5: row 2 (“exsitence”) and row 6 (“filed”).***
>
> **A5**: All typos are fixed and the suggested works are discussed in the revised paper.
>
> ***Q6: Question 1) Why didn’t the authors utilize DINOv2 [1], which has been shown to exhibit greater object awareness, instead of DINO [2]?***
>
> **A6**: For clarification, the VoteCut method used in CuVLER actually has already utilized both DINO and DINOv2 features to obtain rough object masks. In particular, VoteCut uses pre-trained features from four DINO models (DINO\_b8, DINO\_s8, DINO\_b16, DINO\_s16) and two DINOv2 models (DINOv2\_s14, DINOv2\_b14) to get high-quality object masks by voting from the six models together.
>
> Nevertheless, as requested by the reviewer, we further validate the effectiveness of pure DINOv2 features. Particularly, we only use the total four DINOv2 models to generate rough masks on ImageNet, and then train our objectness network on these rough masks, followed by multi-object reasoning on the COCO\* validation set. The following Table 3 shows the results of our direct object discovery. We can see that only using the total four DINOv2 models leads to worse results compared with the original setting of VoteCut. The primary reason is that DINOv2 features are derived from much larger image patches, i.e., $14\times 14$, while DINO features can be from smaller patches, i.e., $8\times 8$. This means the obtained rough object masks from all DINOv2 features tend to be much coarser, resulting in the trained objectness network being inferior and the final performance drop.
>
> *Table 3: The effectiveness of all DINOv2 models for direct object discovery on COCO\* validation set. The first row uses DINO\_b8, DINO\_s8, DINO\_b16, DINO\_s16, DINOv2\_s14, DINOv2\_b14 as described in VoteCut. The second row uses the total four DINOv2 models DINOv2\_s14, DINOv2\_b14, DINO\_l14, DINO\_g14.*
> |  | $\mathrm{AP}_{50}^{\text {box}}$ | $\mathrm{AP}_{75}^{\text {box}}$ | $\mathrm{AP}^{\text {box}}$ | $\mathrm{AR}_{100}^{\text {box}}$ | $\mathrm{AP}_{50}^{\text {mask}}$ | $\mathrm{AP}_{75}^{\text {mask}}$ | $\mathrm{AP}^{\text {mask}}$ | $\mathrm{AR}_{100}^{\text {mask}}$ |
> |---|---|---|---|---|---|---|---|---|
> | 4 DINO models + 2 DINOv2 models | 19.1 | 9.0 | 10.1 | 19.6 | 17.8 | 8.7 | 9.5 | 18.9 |
> | 4 DINOv2 models | 12.0 | 4.6 | 5.2 | 16.8| 12.8 | 3.6 | 5.0 | 16.6 |
>
> ***Q7: Question 2) On average, how many iterations does it take to form the final bounding box?***
>
> **A7**: In the implementation, the maximum number of iterations to optimize a proposal is set as 50. Nevertheless, in practice, as shown in Figure 19 of Appendix A.14, we observe that all proposals tend to converge after just 10 iterations. Therefore, our multi-object reasoning module is generally efficient.

---

> > ### Comment · Reviewer_gBo7 · 2024-11-25
> >
> > Thank you for the response and the additional experiments. I have carefully reviewed the authors’ rebuttal as well as the other reviews. Here are my comments:
> > * My primary concern remains the reliance on additional information gathering in the first step. If fairness is grounded in external human labels, then all models should be considered comparable. However, they are not, as additional training using better pseudo labels leads to improvements. The key difference lies in the presence of a learnable module, whether it is an explicit detector or not. As long as the module is trained to identify features related to objects, it represents a significant modular distinction. Methods like VoteCut and MaskCut can discover objects without training, qualifying as zero-shot detection using DINO. Treating OCN as a form of $g'(dino(\mathbb{I}))$ is problematic, as $g(dino(\mathbb{I}))$ involves pseudo-labeling with no trainable components and is zero-shot, whereas $g'(dino(\mathbb{I}))$ is not. Considering it with an additional step, $f(g(dino(\mathbb{I})))$ better reflects the distinction.
> > * I appreciate the effort in providing analyses for the other experiments, such as RPN, random cropping, and DINOv2. The experiments show that RPN initialization versus random initialization does not result in significant differences, DINOv2 faces precision issues due to patch size, and random cropping brings a notable improvement.
> >
> > Given my primary concern and the points raised about novelty and similarity to prior work (as noted by other reviewers), I will maintain my initial rating.

---

> > > ### Author Response · Authors · 2024-11-28
> > > **Responses to Comments from Reviewer gBo7 (Part 1)**
> > >
> > > **Comment \#1: My primary concern remains the reliance on additional information gathering in the first step. If fairness is grounded in external human labels, then all models should be considered comparable.**
> > >
> > > **Response \#1:**
> > >
> > > (1) Thank you for acknowledging that all methods are fair to compare as long as no human annotations are used. Indeed, this is a de facto standard in the field of unsupervised object segmentation. Here is concrete evidence,
> > >
> > > - In Table 1 of unSAM(NeurIPS'24)[1], all unSAM models are trained based on the inference results of CutLER(CVPR'23)[3] using additional Mask2Former networks, but unSAM models make a comparison with CutLER.
> > > -  In Tables 1/2/3/4 of CuVLER (CVPR'24)[2], it compares with CutLER in all experiments, but it utilizes 5 more DINO/v2 models than CutLER.
> > > - In Table 3 of CutLER, the baselines DINO (ICCV'21)[16] and TokenCut (CVPR'22)[4] do not use additional detectors. But they are compared with CutLER in a single table.
> > > - In Table 1 of TokenCut[4], the baselines (Selective Search[10], EdgeBox[11], Kim et al.[12], Zhang et al.[13], DDT+[14], and rOSD[15]) do not use self-supervised DINO features, but they are compared with TokenCut which uses DINO features.
> > > - In Tables 1/4 of LOST(BMVC'21)[5], the baselines Selective Search[10] or EdgeBox[11] do not involve deep learning features or models, but they are compared with LOST which uses DINO features.
> > >
> > >
> > > In addition, we would bring attention to the fact that, in these related papers, it is a common practice to separately evaluate on two settings: "direct object discovery" and "direct object discovery + detector". For example:
> > >
> > > - In Table 1 of CuVLER[2], the performance without and with Class Agnostic Detector (CAD) is reported.
> > > - In Table 6 of CutLER[3], the results of MaskCut (without CAD) and CutLER (with CAD) are reported
> > > - In Table 1 of TokenCut[4], results without and with CAD are reported.
> > > - In Tables 1/4 of LOST[5], results without and with CAD are separately reported.
> > >
> > > **References:**
> > >
> > > [1] Wang, Xudong, Jingfeng Yang, and Trevor Darrell. "Segment anything without supervision." NeurIPS, 2024.
> > >
> > > [2] Arica, Shahaf, et al. "CuVLER: Enhanced Unsupervised Object Discoveries through Exhaustive Self-Supervised Transformers." CVPR, 2024.
> > >
> > > [3] Wang, Xudong, et al. "Cut and learn for unsupervised object detection and instance segmentation."  CVPR, 2023.
> > >
> > > [4] Wang, Yangtao, et al. "Self-supervised transformers for unsupervised object discovery using normalized cut." CVPR, 2022.
> > >
> > > [5] Siméoni, Oriane, et al. "Localizing objects with self-supervised transformers and no labels." BMVC, 2021.
> > >
> > > [6] Gall, Juergen, and Victor Lempitsky. "Class-specific hough forests for object detection."  CVPR, 2009.
> > >
> > > [7] Gall, Juergen, et al. "Hough forests for object detection, tracking, and action recognition." TPAMI, 2011.
> > >
> > > [8] Qi, Charles R., et al. "Deep hough voting for 3d object detection in point clouds." ICCV, 2019.
> > >
> > > [9] Park, Jeong Joon, et al. "Deepsdf: Learning continuous signed distance functions for shape representation." CVPR, 2019.
> > >
> > > [10] Uijlings, Jasper RR, et al. "Selective search for object recognition." IJCV, 2013.
> > >
> > > [11] Zitnick, C. Lawrence, and Piotr Dollár. "Edge boxes: Locating object proposals from edges." ECCV, 2014.
> > >
> > > [12] Kim, Gunhee, and Antonio Torralba. "Unsupervised detection of regions of interest using iterative link analysis." NeurIPS, 2009.
> > >
> > > [13] Zhang, Runsheng, et al. "Object discovery from a single unlabeled image by mining frequent itemsets with multi-scale features." TIP, 2020.
> > >
> > > [14] Wei, Xiu-Shen, et al. "Unsupervised object discovery and co-localization by deep descriptor transformation." Pattern Recognition, 2019.
> > >
> > > [15] Vo, Huy V., Patrick Pérez, and Jean Ponce. "Toward unsupervised, multi-object discovery in large-scale image collections." ECCV, 2020.
> > >
> > > [16] Caron, Mathilde, et al. "Emerging properties in self-supervised vision transformers." ICCV, 2021.

---

> > > ### Author Response · Authors · 2024-11-28
> > > **Responses to Comments from Reviewer gBo7 (Part 2)**
> > >
> > > **Comment \#2: However, they are not, as additional training using better pseudo labels leads to improvements. The key difference lies in the presence of a learnable module, whether it is an explicit detector or not. As long as the module is trained to identify features related to objects, it represents a significant modular distinction. Methods like VoteCut and MaskCut can discover objects without training, qualifying as zero-shot detection using DINO. Treating OCN as a form of $g'(dino(\mathbb{I}))$ is problematic, as $g(dino(\mathbb{I}))$ involves pseudo-labeling with no trainable components and is zero-shot, whereas $g'(dino(\mathbb{I}))$ is not. Considering it with an additional step, $f(g(dino(\mathbb{I})))$ better reflects the distinction.**
> > >
> > > **Response \#2:** We respect the reviewer's opinion, and we are happy to: either **1) merge Table 1** of Section 4.1 "Direct Object Discovery" with Table 2 of Section 4.2 "Training a Detector", or **2) move Table 1** of Section 4.1 to Appendix, or **3) remove Section 4.1 straight away**. We are kindly awaiting your further suggestion.
> > >
> > > Regarding Table 2 of our Section 4.2, we compare with the baselines of CutLER, CuVLER and unSAM, all based on trained additional detectors using their own discovered objects via various means. Following the reviewer's interests in trainable networks, we make the summary as follows:
> > >
> > > - CutLER trains 3 additional detectors $f_3/f_2/f_1$ on pseudo-labels generated by MaskCut on ImageNet, *i.e.*, $f_3(f_2(f_1(MaskCut(\mathbb{I}))))$.
> > > - unSAM trains 4 additional detectors $f_4/f_3/f_2/f_1$, on pseudo-labels generated by MaskCut on ImageNet, *i.e.*, $f_4(f_3(f_2(f_1(MaskCut(\mathbb{I})))))$.
> > > - CuVLER trains 2 additional detector $f_2/f_1$ on pseudo-labels generated by VoteCut on ImageNet, *i.e.*, $f_2(f_1(VoteCut(\mathbb{I})))$.
> > > - Our OCN trains one objectness network $f_1$ and one additional detector $f_2$ on pseudo-labels generated by VoteCut on ImageNet,  *i.e.* , $f_2(f_1(VoteCut(\mathbb{I})))$.
> > >
> > >
> > > This means that, in terms of trainable networks, our OCN is strictly comparable with CuVLER, whereas CutLER and unSAM rely on more learnable components to identify features related to objects.
> > >
> > > **Comment \#3: I appreciate the effort in providing analyses for the other experiments, such as RPN, random cropping, and DINOv2. The experiments show that RPN initialization versus random initialization does not result in significant differences, DINOv2 faces precision issues due to patch size, and random cropping brings a notable improvement.**
> > >
> > > **Response \#3:** Thank you for acknowledging our rebuttal materials and we are glad that your concerns have been addressed accordingly.
> > >
> > >
> > > **Comment \#4: Given my primary concern and the points raised about novelty and similarity to prior work (as noted by other reviewers), I will maintain my initial rating.**
> > >
> > > **Response \#4:** Regarding your primary concern about the comparison in Table 1 of Section 4.1 "Direct Object Discovery", refer to our clarifications in Responses \#1 and \#2. We are kindly awaiting your further suggestion.
> > >
> > > Regarding our novelty, kindly refer to our concrete Response \#1 to the reviewer **UESN**. To sum up, the reviewer **UESN** constantly criticizes our novelty, but without providing any related publications so far. We are always glad to discuss further if you are willing to share related works as well.

---

> ### Author Response · Authors · 2024-11-30
> **Responses to Comments from Reviewer gBo7**
>
> **Comment \#1: Comparison in Table 1: The issue with Table 1 lies in comparing models not to the full models but to pseudo-labeling approaches of CuVLER (VoteCut) and CutLER (MaskCut). Essentially, OCN is compared to its pseudo-label outputs, specifically VoteCut. The models compared in Table 1 include both learning-based approaches (DINOSAUR, FOUND) and non-learning-based ones (FreeMask, MaskCut, VoteCut). This distinction is presented clearly in Table 1 of the FOUND paper. The most tangible criterion for categorization seems to be the presence of learnable parameters rather than the inclusion of a detector.**
>
> **Response \#1:** Thank you for the concrete comment and sharing the example of Table 1 from FOUND. As suggested, we take the learnable parameters into account for categorization as shown in the following newly organized Table 1.
>
> **Comment \#2: Recommendation for Experimental Organization: To address this issue and ensure fair comparisons, I suggest reorganizing the experiments section. A more comprehensive table could combine the results of Table 1 and Table 2, categorizing the models or, at the very least, clearly indicating the differences based on i) whether they include learnable parameters; ii) the type of learnable module, such as an additional detector. This approach would mitigate the potential ambiguity in defining direct object discovery.**
>
> **Response \#2:** This is a very neat suggestion. We combine our results of the original Tables 1/2 into a single Table 1 as shown below, clearly indicating the differences based on: 1) whether including learnable parameters or not, and 2) the type of learnable modules. This newly organized Table 1 will be updated to the next version.
>
> **Comment \#3: Minor Note on Table 2: For improved clarity and conciseness, Table 2 could be simplified by reporting only the best version of each module from previous work, while retaining the full table in the appendix for reference.**
>
> **Response \#3:** Sure. Only the best result of each method is presented in our newly organized Table 1 as shown below. Other details will be moved to Appendix in the next version.
>
> **Comment \#4: Acknowledgment and Final Score: I thank the authors for their effort and dedication in addressing the reviewers’ concerns. If the fairness of the experimental setup is adequately resolved, I am inclined to raise my score to acceptance.**
>
> **Response \#4:** We highly appreciate the reviewer's all valuable suggestions. The experimental setup of the following newly organized Table 1 is now clearly fair.
>
>
> *Table 1: Quantitative results on COCO\* validation dataset. CAD is short for Class Agnostic Detector.*
> |  | Trainable Module | Detector Setting | $\text{AP}^{\text{box}}_{\text{50}}$ | $\text{AP}^{\text{box}}_{\text{75}}$ | $\text{AP}^{\text{box}}$ | $\text{AR}^{\text{box}}_{\text{100}}$ | $\text{AP}^{\text{mask}}_{\text{50}}$ | $\text{AP}^{\text{mask}}_{\text{75}}$ | $\text{AP}^{\text{mask}}$ | $\text{AR}^{\text{mask}}_{\text{100}}$ |
> |---|:---:|:---:|:---:|:---:|:---:|:---:|:---:|:---:|:---:|:---:|
> |  |  |  |  | **---** | **without** | **learning** | **---** |  |  |  |
> | FreeMask | - | - | 3.7 | 0.6 | 1.3 | 4.6 | 3.1 | 0.3 | 0.9 | 3.5 |
> | MaskCut (K=3) | - | - | 6.0 | 2.4 | 2.9 | 6.7 | 5.1 | 1.8 | 2.3 | 5.8 |
> | MaskCut (K=10) | - | - | 6.2 | 2.6 | 2.9 | 7.2 | 5.3 | 2.0 | 2.3 | 6.2 |
> | VoteCut | - | - | 10.8 | 4.9 | 5.5 | 11.3 | 9.5 | 4.0 | 4.6 | 9.8 |
> |  |  |  |  | **---** | **with** | **learning** | **---** |  |  |  |
> | **without CAD:** | |  |  |  |  |  |  |  |  |  |
> | DINOSAUR | Recon. SlotAtt | - | 2.0 | 0.2 | 0.6 | 4.8 | 1.1 | 0.1 | 0.3 | 2.9 |
> | FOUND | Seg. Head | - | 4.4 | 1.8 | 2.1 | 3.6 | 3.3 | 1.3 | 1.5 | 3.0 |
> | OCN$_{disc}$ | Obj. Net | - | 19.1 | 9.0 | 10.1 | 19.6 | 17.8 | 8.7 | 9.5 | 18.9 |
> | **with CAD:** | |  |  |  |  |  |  |  |  |  |
> | unSAM | Detector x 4 | setting #2 | 10.2 | 6.3 | 6.4 | 36.1 | 10.2 | 6.2 | 6.3 | 34.1 |
> | CutLER | Detector x 3 | setting #3 | 26.0 | 14.2 | 14.7 | 37.9 | 22.7 | 11.2 | 11.8 | 32.7 |
> | CuVLER | Detector x 2 | setting #4 | 28.0 | 14.8 | 15.5 | 37.8 | 24.4 | 11.7 | 12.6 | 32.1 |
> | **OCN** | **Obj. Network   + Detector x 1** | **setting #2** | **32.6** | **17.2** | **18.0** | **40.9** | **29.6** | **14.4** | **15.5** | **36.5** |

---

> > ### Comment · Reviewer_gBo7 · 2024-12-02
> > **Final Score**
> >
> > My concerns regarding the fairness of the evaluation have been resolved. Therefore, I am increasing my score.

---

> > > ### Author Response · Authors · 2024-12-02
> > > **Thanks**
> > >
> > > Dear reviewer gBo7,
> > >
> > > We appreciate your precious time and positive rating on our paper.  Your insightful suggestions have significantly improved our manuscript.
> > >
> > > Best,
> > > Authors

---

### Official Review · Reviewer_ffrn · 2024-10-30

**Soundness:** 3
**Presentation:** 3
**Contribution:** 3
**Rating:** 8
**Confidence:** 2

**Summary:**

This paper studies the challenging problem: unsupervised multi-object segmentation. Previous methods based on slot-attention usually fell short on complex scenarios such as COCO, while self-supervised feature distillation methods usually fail to discover multi-objects. This paper proposes a two-stage framework, including object-centric representation learning and multi-object reasoning. For the first stage, the authors first explicitly identify the object existence score, the object center field, and the object boundary distance field, as the representation, and then train an objectiveness network to learn this type of representation on ImageNet. For the second stage, an iterative algorithm is applied. Experiments under various evaluation protocols demonstrate the effectiveness of the proposed method.

**Strengths:**

- The topic is challenging and worth studying.
- This paper is well-written and easy to follow.
- The figures have vividly illustrated the proposed method.
- The proposed method is quite effective.
- The defined explicit object-centric representation is reasonable.

**Weaknesses:**

I only have one concern: the latency. The proposed method needs to iteratively leverage the objectiveness network, which is not that efficient. Could you compare the evaluation cost?

**Questions:**

I have no further questions.

---

> ### Author Response · Authors · 2024-11-24
> **Responses to Reviewer ffrn**
>
> We appreciate the reviewer’s very positive comments and address the main concern below.
>
> **Q1: I only have one concern: the latency. The proposed method needs to iteratively leverage the objectiveness network, which is not that efficient. Could you compare the evaluation cost?**
>
> **A1**: As requested, we compare our method with baselines regarding the efficiency of object segmentation. In particular, we compute the average time consumption to process one multi-object image in two settings: 1) direct object discovery, and 2) inference by a trained multi-object detector.
>
> Table 1 below reports the averaged time on 5000 images from COCO\* validation set. We can see that, for direct object discovery, as expected, our OCN$_{disc}$ takes the longest time to discover all objects due to the iterative optimization manner. Nevertheless, once we train an additional multi-object detector using our discovered pseudo-labels, we just need the same time as baselines to infer multiple objects in a single forward pass.
>
> *Table 1: Inference time cost at different settings.*
> | inference time (seconds per image) |  |  |  |  |
> |:---:|:---:|:---:|:---:|:---:|
> |  | MaskCut (N=3) | MaskCut (N=10) | VoteCut | OCN$_{disc}$ |
> | Direct Object Discovery | 11.3 | 33.7 | 5.1 | 45.3 |
> |  | CutLER | CutLER | CuVLER | OCN |
> | Inference with Detector | 0.1 | 0.1 | 0.1 | 0.1 |

---

> > ### Comment · Reviewer_ffrn · 2024-11-25
> > **Post Rebuttal Comments**
> >
> > I appreciate the efficiency comparison, where OCN is as efficient as other methods after training an additional detector. Therefore, I will keep my initial rating and recommend acceptance.

---

> > > ### Author Response · Authors · 2024-11-25
> > > **Thanks**
> > >
> > > We appreciate the reviewer's very encouraging feedback.  -Authors

---

### Official Review · Reviewer_UESN · 2024-11-03

**Soundness:** 3
**Presentation:** 3
**Contribution:** 2
**Rating:** 5
**Confidence:** 4

**Summary:**

This paper addresses unsupervised multi-object segmentation by learning an object-centric representation. The proposed method is composed of two stages. In the first stage, an object-centric representation of three levels is derived, encompassing object existence, center, and boundary levels. This representation is class-agnostic, enabling the segmentation of objects from unseen classes. In the subsequent stage, a multi-object reasoning module, built upon the derived representation, identifies object instances. The proposed method is evaluated on multiple datasets in different settings and achieves promising results.

**Strengths:**

1. The paper, in general, is well-written and easy to follow.

2. The proposed method is simple and reasonably designed.

3. Recognizing the limitations of the COCO validation set, the authors annotated object instances that were not originally labeled, significantly enhancing its value for unsupervised object discovery.

4. The proposed method demonstrates superior performance across various datasets and experimental settings. Additionally, ablation studies confirm the effectiveness of each component in the three-level, object-centric representation.

**Weaknesses:**

1. My primary concern regarding this paper is the limited novelty and technical contributions. The proposed method consists of two main components: a) an objectness network for extracting a three-level, object-centric representation and b) a multi-object reasoning module for unsupervised object discovery.

a. The three-level representation, encompassing object existence, center, and boundary information, has been widely explored in the literature. For example, techniques like the Hough transform and Chamfer distance have been extensively used to capture similar representations, where object centers and boundaries are encoded. Using these inherently class-agnostic representations for unsupervised object segmentation is a relatively straightforward extension, limiting the degree of novelty.

b. The multi-object reasoning module, composed of four sequential steps, is designed in a heuristic way. Each step processes the features from individual representation levels, potentially hindering the model's ability to fully exploit the interdependencies between these levels.

2. In general, this paper is clearly written. However, to further enhance its clarity and impact, the following suggestions may be considered:

a. It may be better if Figures 3, 4, and 5 can be integrated into Figure 1. The three-level presentation is repeated several times on pages 1 and 2 with a reference to Figure 1. However, readers may clearly realize what the three-level representation is after seeing the example in Figures 3, 4, and 5 on pages 3 and 4.

b. In Section 2, it would be better to discuss why the proposed method is superior to existing methods, especially those learning object-centric representations with pre-trained features, since the proposed method uses pre-trained features, too.

c. A deeper analysis of the experimental results is necessary. While the paper emphasizes the improved performance of the proposed method, a more in-depth exploration of the underlying reasons for this superiority would strengthen the overall argument.

d. The indexing of the four steps in the reasoning module should be consistent throughout the paper, either using #0 to #3 on page 5 or #1 to #4 on page 6.

3. The sensitivity analysis presented in Table 10 of the supplementary materials is limited in scope. The narrow value ranges of hyperparameter values and the lack of evaluation on multiple datasets hinder a comprehensive assessment of the method's sensitivity to hyperparameter variations across different datasets.

**Questions:**

1. The authors might address my comments given in Weaknesses.

2. Please check the correctness of the statement in Lines 255 ~ 258. Consider a proposal where two objects separated by a distance greater than five pixels are present. Can this proposal be correctly excluded by using the designed kernel in Figure 6?

3. Why are different competing methods adopted in different experiments, namely those in Tables 1 ~ 4?

4. As the images in the COCO* dataset are newly annotated by the authors, did the authors tune the hyperparameters of the competing methods to report the performance of these methods?

---

> ### Author Response · Authors · 2024-11-24
> **Responses to Reviewer UESN (Part 1)**
>
> We appreciate the reviewer's thoughtful comments and address the main concerns below.
>
> ***Q1:  1a) The three-level representation, encompassing object existence, ... limiting the degree of novelty.***
>
> **A1**: Thank you for pointing out related works and we would clarify our novelty as follows:
>
> First, regarding our design of the object center field, we notice that prior works [1,2,3] use Hough Transform to transform pixels/points to object centroids for 2D/3D object detection, which requires learning both directions and distances to object centers. However, our object center field is just defined as unit directions pointing to object centers, as we only need to learn such directions to identify multi-center proposals instead of recovering object masks.
>
> Second, regarding our object boundary distance field, we agree that the concept of boundary distance field is successfully used in literature, but it is mainly used for shape reconstruction [4,5]. In this paper, we demonstrate its effectiveness for object discovery.
>
> Overall, our object center field and boundary distance field are particularly designed for representing objects with sound motivations, and ultimately achieve superior performance for unsupervised object discovery. To the best of our knowledge, there is no other similar work in the field of unsupervised object segmentation, though the related concepts of object centers and boundaries are indeed not entirely new in the broader literature.
>
> In the revised paper, we discuss the related works and highlight the differences with our design in lines 182-186 and lines 216-218 of Section 3.2.
>
> ***Q2: 1b) The multi-object reasoning module, composed of four sequential steps, is designed in a heuristic way. ... these levels.***
>
> **A2**: We agree that our multi-object reasoning module follows a heuristic design. Nevertheless, we would clarify that, because our carefully introduced three level object-centric representations are inherently coupled to jointly describe objectness, our multi-object reasoning module should follow a series of logic steps to gradually infer multiple objects in the absence of any human labels. This is one of our core contributions to enable our whole pipeline to be superior to baselines.
>
> Nevertheless, it may also be possible to leverage reinforcement learning techniques to learn an efficient policy network to update object proposals. We would leave this non-trivial exploration for our future work.
>
> In the revised paper, we briefly clarify this design principle in lines 97-98 of Section 1.
>
> ***Q3: 2a) It may be better if Figures 3, 4, and 5 ca...mple in Figures 3, 4, and 5 on pages 3 and 4.***
>
> **A3**: This is a very nice suggestion and we have updated Figure 1 accordingly in the revised paper.
>
> ***Q4: 2b) In Section 2, it would be better to discuss why the proposed method is superior to existing methods, especially those learning object-centric representations with pre-trained features, since the proposed method uses pre-trained features, too.***
>
> **A4**: Thanks for the advice and we have updated the discussion in lines 139-145 of Section 2 in the revised paper.
>
> Our method is superior to existing methods, essentially because existing methods tend to simply group pixels with similar features (obtained from pretrained models) as a single object, lacking the ability to discern boundaries between objects. As a consequence, for example, they usually group two chairs nearby into just one object. By contrast, our introduced three level object-centric representations are designed to jointly retain unique and explicit objectness features for each pixel, \ie{}, how far away to the object boundary and in what direction to the object center.
>
>
>
> **References**
>
> $[1]$ Gall, Juergen, and Victor Lempitsky. "Class-specific hough forests for object detection." Decision forests for computer vision and medical image analysis (2013): 143-157.
>
> $[2]$ Gall, Juergen, et al. "Hough forests for object detection, tracking, and action recognition." IEEE transactions on pattern analysis and machine intelligence 33.11 (2011): 2188-2202.
>
> $[3]$ Qi, Charles R., et al. "Deep hough voting for 3d object detection in point clouds." proceedings of the IEEE/CVF International Conference on Computer Vision. 2019.
>
> $[4]$ Park, Jeong Joon, et al. "Deepsdf: Learning continuous signed distance functions for shape representation." Proceedings of the IEEE/CVF conference on computer vision and pattern recognition. 2019.
>
> $[5]$ Xie, Yiheng, et al. "Neural fields in visual computing and beyond." Computer Graphics Forum. Vol. 41. No. 2. 2022.

---

> > ### Author Response · Authors · 2024-11-24
> > **Responses to Reviewer UESN (Part 5)**
> >
> > ***Q8: Questions 2) Please check the correctness of the statement in Lines 255 - 258. Consider a proposal where two objects separated by a distance greater than five pixels are present. Can this proposal be correctly excluded by using the designed kernel in Figure 6?***
> >
> > **A8**: Thank you for pointing out this flaw. We agree that when multiple (sparse) objects within the same proposal are separated by more than 5 pixels, our original design of center reasoning would ignore it.
> >
> > In the revised paper, we address this issue as follows. In Step \#2 of Section 3.3, if the highest value of $\boldsymbol{f}_p^{ac}$ is smaller than the threshold $\tau^c$, the proposal $P$ is likely to have just one object, or multiple objects but they are far away from each other, \ie{}, more than 5 pixels apart. In this regard, we simply adopt the connected-component method used in CuVLER to split the proposal $P$ into subproposals. Particularly, for its center field $\boldsymbol{f}^c_p$, all pixels that are spatially connected and have non-zero unit vectors are grouped into one subproposal. Each subproposal is regarded as a brand-new one and will be evaluated from Step \#1 again.
> >
> > Nevertheless, after adding the above strategy, we empirically find that it has little effect on the final results of our direct object discovery, as shown in the following table. Upon a closer investigation, we find that, for images with multiple but sparsely located objects, it is actually very easy and fast to accurately discover those individual objects. Basically, this is because we have thousands of proposals randomly initialized with various sizes, and there are always good chances to capture those isolated objects.
> >
> > In the revised paper, to ensure the completeness of our method, we choose to add the connected-component solution to address the issue, as outlined in lines 283-289 of Section 3.3.
> >
> > |  | $\mathrm{AP}_{50}^{\text {box}}$ | $\mathrm{AP}_{75}^{\text {box}}$ | $\mathrm{AP}^{\text {box}}$ | $\mathrm{AR}_{100}^{\text {box}}$ | $\mathrm{AP}_{50}^{\text {mask}}$ | $\mathrm{AP}_{75}^{\text {mask}}$ | $\mathrm{AP}^{\text {mask}}$ | $\mathrm{AR}_{100}^{\text {mask}}$ |
> > |---|---|---|---|---|---|---|---|---|
> > | $OCN_{disc}$ | 19.1 | 9.0 | 10.1 | 19.6 | 17.8 | 8.7 | 9.5 | 18.9 |
> > | $OCN_{disc}$ (+CC) | 19.1 | 9.0 | 10.1 | 19.6 | 17.8 | 8.7 | 9.5 | 18.9 |
> >
> > ***Q9: Questions 3) Why are different competing methods adopted in different experiments, namely those in Tables 1 - 4?***
> >
> >
> > **A9**: This confusion may be caused by the unclear method acronym assigned in original Tables 1-4.
> >
> > Following prior works CutLER/ CuVLER for comprehensive comparisons, our main paper has three evaluation protocols: 1) Direct Object Discovery, 2) Training an additional Detector, and 3) Zero-shot Detection with the Detector. In each protocol, we need to compare with different baselines because of the different settings applied.
> >
> > To be clear, in the first protocol, we name our method OCN$_{disc}$ in Table 1, whereas we name our method OCN in the second and third protocols. Note that, due to the space limitation, the original Table 2 has been moved to Table 6 of the Appendix in the revised paper.
> >
> > ***Q10: Questions 4) As the images in the COCO\* dataset are newly annotated by the authors, did the authors tune the hyperparameters of the competing methods to report the performance of these methods?***
> >
> > **A10**: For the baseline MaskCut, we apply two choices (3 and 10) for its hyperparameter $K$, though both achieve nearly the same performance. For other baselines, there are no other hyperparameters to tune. Notably, our newly annotated COCO\* val set is held out for a purely unsupervised evaluation, not for hyperparameter tuning for any methods including ours.

---

> ### Author Response · Authors · 2024-11-24
> **Responses to Reviewer UESN (Part 2)**
>
> ***Q5: 2c) A deeper analysis of the experimental results is necessary. While the paper emphasizes the improved performance of the proposed method, a more in-depth exploration of the underlying reasons for this superiority would strengthen the overall argument.***
>
> **A5**: We appreciate this valuable suggestion. In the revised paper, we have updated Table 1 by adding more in-depth quantitative results such as Precision/Recall scores, and updated Figure 5 by visualizing the DINO/v2 features used by baselines for discovering objects. In addition, we have added a new paragraph to analyze our results in lines 391-402 of Section 4.1. The analysis is as follows:
>
>  From Table 1 (also shown at Table 1 below), we can see that the baselines such as FOUND and MaskCut can achieve high precision scores, but have rather low recall scores, meaning that they tend to correctly discover just a few objects. By contrast, our method for direct object discovery, named OCN$_{disc}$, achieves balanced precision and recall scores, meaning that we can correctly discover many more objects. Fundamentally, this is because the baselines mainly rely on grouping similar patch features (obtained from pretrained DINO/v2) as objects, resulting in multiple similar objects being grouped as just one, as shown in Figure 5 where two cabinets are detected as one. However, our method learns clear object centers and boundaries, allowing us to easily discover individual objects, especially on crowded scenes.
>
> *Table 1: Quantitative results of direct object discovery on COCO\* validation set.*
> |  | $\mathrm{AP}_{50}^{\text {box}}$ | $\mathrm{AP}_{75}^{\text {box}}$ | $\mathrm{AP}^{\text {box}}$ | $\mathrm{AR}_{100}^{\text {box}}$ | $\mathrm{AP}_{50}^{\text {mask}}$ | $\mathrm{AP}_{75}^{\text {mask}}$ | $\mathrm{AP}^{\text {mask}}$ | $\mathrm{AR}_{100}^{\text {mask}}$ | $\mathrm{Pre}_{50}^{\text {mask}}$ | $\mathrm{Rec}_{50}^{\text {mask}}$ | $\mathrm{Pre}_{75}^{\text {mask}}$ | $\mathrm{Rec}_{75}^{\text {mask}}$ |
> |---:|---|---|---|---|---|---|---|---|---|---|---|---|
> | DINOSAUR | 2.0 | 0.2 | 0.6 | 4.8 | 1.1 | 0.1 | 0.3 | 2.9 | 13.1 | 10.0 | 3.0 | 2.2 |
> | FOUND | 4.4 | 1.8 | 2.1 | 3.6 | 3.3 | 1.3 | 1.5 | 3.0 | **51.1** | 5.5 | 26.9 | 2.9 |
> | FreeMask | 3.7 | 0.6 | 1.3 | 4.6 | 3.1 | 0.3 | 0.9 | 3.5 | 22.8 | 9.1 | 5.3 | 2.1 |
> | MaskCut(K=3) | 6.0 | 2.4 | 2.9 | 6.7 | 5.1 | 1.8 | 2.3 | 5.8 | 50.4 | 10.1 | **30.0** | 5.7 |
> | MaskCut(K=10) | 6.2 | 2.6 | 2.9 | 7.2 | 5.3 | 2.0 | 2.3 | 6.2 | 48.0 | 10.9 | 27.3 | 6.1 |
> | VoteCut | 10.8 | 4.9 | 5.5 | 11.3 | 9.5 | 4.0 | 4.6 | 9.8 | 21.0 | 17.2 | 10.6 | 9.7 |
> | **OCN$_{\text {disc }}$** | **19.1** | **9.0** | **10.1** | **19.6** | **17.8** | **8.7** | **9.5** | **18.9** | 35.5 | **30.0** | 22.1 | **19.6** |
>
>
> ***Q6: 2d) The indexing of the four steps in the reasoning module should be consistent throughout the paper, either using \#0 to \#3 on page 5 or \#1 to \#4 on page 6.***
>
> **A6**: The typos have been fixed in the revised paper.
>
> ***Q7: 3) The sensitivity analysis presented in Table 10 of the supplementary materials is limited in scope. The narrow value ranges of hyperparameter values and the lack of evaluation on multiple datasets hinder a comprehensive assessment of the method's sensitivity to hyperparameter variations across different datasets.***
>
> **A7**: As requested, in the revised paper (Tables 10 and 11), we conducted more ablation experiments on a wide range of $(0\sim 0.95)$ for all three hyperparameters $\tau^e_{conf} / \tau^c_{conf}$ / $\tau^b_{conf}$ on all 7 datasets. The new results are also provided in the following two tables (Table 2 and Table 3 in Response Part 3 and Part 4).
>
> We can see that more tolerant thresholds lead to higher AR scores because more objects can be discovered, but a decrease in AP because of low-quality detections. On the other hand, if thresholds are too strict, both AR and AP scores drop because only a limited number of objects are discovered.
> Nevertheless, our method is not particularly sensitive to the selection of thresholds as it demonstrates good performance across different thresholds.

---

> ### Author Response · Authors · 2024-11-24
> **Responses to Reviewer UESN (Part 3)**
>
> *Table 2: Ablation results for threshold of object existence $\tau^e_{conf}$, object center $\tau^c_{conf}$ and object boundary $\tau^b_{conf}$ on COCO\* validation set.*
> | $\tau^{e}_{conf}$ | $\tau^{c}_{conf}$ | $\tau^{b}_{conf}$ | $\mathrm{AP}_{50}^{\text {box}}$ | $\mathrm{AP}_{75}^{\text {box}}$ | $\mathrm{AP}^{\text {box}}$ | $\mathrm{AR}_{100}^{\text {box}}$ | $\mathrm{AP}_{50}^{\text {mask}}$ | $\mathrm{AP}_{75}^{\text {mask}}$ | $\mathrm{AP}^{\text {mask}}$ | $\mathrm{AR}_{100}^{\text {mask}}$ |
> |---|---|---|---|---|---|---|---|---|---|---|
> | **0.0** | 0.8 | 0.75 | 31.2 | 16.7 | 17.4 | **41.0** | 28.7 | **14.6** | 15.3 | **37.2** |
> | **0.25** | 0.8 | 0.75 | 31.5 | 16.7 | 17.5 | 40.8 | 28.6 | 14.3 | 15.2 | 36.7 |
> | **_0.5_** | 0.8 | 0.75 | **32.6** | **17.2** | **18.0** | 40.9 | **29.6** | 14.4 | **15.5** | 36.5 |
> | **0.75** | 0.8 | 0.75 | 30.8 | 16.2 | 16.9 | 38.9 | 27.7 | 13.3 | 14.3 | 34.7 |
> | **0.95** | 0.8 | 0.75 | 28.1 | 13.4 | 14.7 | 34.4 | 24.3 | 10.7 | 12.1 | 30.1 |
> | 0.5 | **0.0** | 0.75 | 32.5 | 16.4 | 17.5 | 40.0 | 29.2 | 13.6 | 14.9 | 35.8 |
> | 0.5 | **0.25** | 0.75 | 31.8 | 16.4 | 17.3 | 39.9 | 28.5 | 13.5 | 14.7 | 35.7 |
> | 0.5 | **0.5** | 0.75 | 31.0 | 16.2 | 17.0 | 40.2 | 27.7 | 13.3 | 14.4 | 36.0 |
> | 0.5 | **_0.8_** | 0.75 | **32.6** | **17.2** | **18.0** | **40.9** | **29.6** | **14.4** | **15.5** | **36.5** |
> | 0.5 | **0.95** | 0.75 | 29.8 | 15.8 | 16.5 | 38.1 | 26.8 | 13.2 | 14.1 | 34.2 |
> | 0.5 | 0.8 | **0.0** | 31.8 | 16.0 | 17.0 | 38.7 | 28.4 | 13.2 | 14.5 | 34.6 |
> | 0.5 | 0.8 | **0.25** | 31.2 | 16.1 | 17.0 | 38.9 | 27.8 | 13.2 | 14.3 | 34.7 |
> | 0.5 | 0.8 | **0.5** | 31.7 | 16.9 | 17.5 | 40.6 | 28.4 | 13.7 | 14.7 | 36.0 |
> | 0.5 | 0.8 | **_0.75_** | **32.6** | **17.2** | **18.0** | **40.9** | **29.6** | **14.4** | **15.5** | **36.5** |
> | 0.5 | 0.8 | **0.95** | 31.6 | 17.5 | 17.9 | 39.8 | 28.0 | 13.3 | 14.5 | 35.0 |

---

> ### Author Response · Authors · 2024-11-24
> **Responses to Reviewer UESN (Part 4)**
>
> *Table 3: Ablation results for threshold of object existence $\tau^e_{conf}$, object center $\tau^c_{conf}$ and object boundary $\tau^b_{conf}$ on COCO20K, LVIS, KITTI, VOC, Object365 and OpenImages.*
> |  |  |  | COCO |  |  |  | COCO20K |  |  |  | LVIS |  |  |  | KITTI |  | VOC |  | Object365 |  | OpenImages |  |
> |---|---|---|:---:|:---:|:---:|:---:|:---:|:---:|:---:|:---:|:---:|:---:|:---:|:---:|:---:|:---:|:---:|:---:|:---:|:---:|:---:|:---:|
> | $\tau^{e}_{conf}$ | $\tau^{c}_{conf}$ | $\tau^{b}_{conf}$ | $\mathrm{AP}_{50}^{\text {box}}$  | $\mathrm{AR}_{100}^{\text {box}}$  | $\mathrm{AP}_{50}^{\text {mask}}$  | $\mathrm{AR}_{100}^{\text {mask}}$ | $\mathrm{AP}_{50}^{\text {box}}$  | $\mathrm{AR}_{100}^{\text {box}}$  | $\mathrm{AP}_{50}^{\text {mask}}$  | $\mathrm{AR}_{100}^{\text {mask}}$ | $\mathrm{AP}_{50}^{\text {box}}$  | $\mathrm{AR}_{100}^{\text {box}}$  | $\mathrm{AP}_{50}^{\text {mask}}$  | $\mathrm{AR}_{100}^{\text {mask}}$ | $\mathrm{AP}_{50}^{\text {box}}$  | $\mathrm{AR}_{100}^{\text {box}}$  | $\mathrm{AP}_{50}^{\text {box}}$  | $\mathrm{AR}_{100}^{\text {box}}$  | $\mathrm{AP}_{50}^{\text {box}}$  | $\mathrm{AR}_{100}^{\text {box}}$ | $\mathrm{AP}_{50}^{\text {box}}$  | $\mathrm{AR}_{100}^{\text {box}}$ |
> | **0.0** | 0.8 | 0.75 | 23.8 | 35.1 | 21.9 | **30.8** | 24.3 | 35.2 | 22.6 | **31.1** | 10.2 | **24.9** | **9.0** | **22.6** | 25.3 | 32.5 | 38.5 | 46.9 | 23.6 | **36.3** | 18.3 | **29.5** |
> | **0.25** | 0.8 | 0.75 | 24.1 | 34.8 | 22.0 | 30.3 | 24.6 | 35.0 | 22.6 | 30.6 | 10.2 | 24.4 | 8.7 | 21.9 | 25.0 | 34.0 | 39.1 | 46.6 | 23.8 | 36.0 | 18.7 | 29.4 |
> | **_0.5_** | 0.8 | 0.75 | **25.4** | **35.2** | **22.9** | 30.3 | **25.9** | **35.4** | **23.6** | 30.5 | **10.4** | 24.1 | 8.9 | 21.4 | **26.7** | **34.8** | **40.4** | **47.4** | **24.7** | 35.9 | **19.0** | **29.5** |
> | **0.75** | 0.8 | 0.75 | 24.5 | 33.7 | 21.9 | 28.8 | 25.1 | 34.1 | 22.7 | 29.2 | 9.9 | 22.5 | 8.3 | 20.0 | 25.5 | 33.6 | **40.4** | 46.7 | 23.8 | 36.0 | 18.7 | 29.4 |
> | **0.95** | 0.8 | 0.75 | 23.2 | 30.2 | 19.9 | 25.0 | 23.8 | 30.5 | 20.6 | 25.3 | 8.7 | 18.8 | 6.9 | 16.3 | 21.6 | 29.6 | 39.4 | 43.7 | 21.6 | 30.0 | 18.8 | 26.5 |
> | 0.5 | **0.0** | 0.75 | **25.7** | 34.5 | 22.8 | 29.8 | **26.2** | 34.8 | 23.4 | 30.1 | 10.4 | 23.3 | 8.5 | 20.9 | **28.7** | **35.5** | **41.3** | 47.0 | 24.5 | 35.1 | 19.7 | 29.0 |
> | 0.5 | **0.25** | 0.75 | 25.0 | 34.4 | 22.2 | 29.5 | 25.6 | 34.8 | 23.0 | 29.8 | 10.1 | 23.2 | 8.3 | 20.6 | 27.7 | 33.6 | 41.0 | 46.8 | 23.8 | 35.1 | 19.3 | 29.0 |
> | 0.5 | **0.5** | 0.75 | 24.5 | 34.7 | 21.8 | 29.9 | 25.1 | 34.8 | 22.5 | 30.1 | 9.8 | 23.6 | 8.0 | 21.1 | 24.1 | 32.7 | 40.3 | 46.7 | 23.3 | 35.3 | **19.9** | **29.7** |
> | 0.5 | **_0.8_** | 0.75 | 25.4 | **35.2** | **22.9** | **30.3** | 25.9 | **35.4** | **23.6** | **30.5** | **10.4** | **24.1** | **8.9** | **21.4** | 26.7 | 34.8 | 40.4 | **47.4** | **24.7** | **35.9** | 19.0 | 29.5 |
> | 0.5 | **0.95** | 0.75 | 23.7 | 32.9 | 21.1 | 28.3 | 24.3 | 33.2 | 21.8 | 28.5 | 9.6 | 21.6 | 8.2 | 19.3 | 25.7 | 33.3 | 38.6 | 45.6 | 22.5 | 33.2 | 18.3 | 28.4 |
> | 0.5 | 0.8 | **0.0** | 24.7 | 33.4 | 21.9 | 28.7 | 25.3 | 33.6 | 22.6 | 29.0 | 10.1 | 22.3 | 8.2 | 19.8 | 27.4 | 33.4 | 40.0 | 45.9 | 23.6 | 33.8 | 19.3 | 28.3 |
> | 0.5 | 0.8 | **0.25** | 24.6 | 33.6 | 21.8 | 28.9 | 25.3 | 34.0 | 22.5 | 29.3 | 9.8 | 22.4 | 8.0 | 19.8 | 26.7 | 33.5 | 40.7 | 46.1 | 23.2 | 34.1 | 19.7 | 28.6 |
> | 0.5 | 0.8 | **0.5** | 25.3 | **35.2** | 22.4 | 30.0 | **25.9** | 35.3 | 23.1 | 30.4 | 10.0 | 23.6 | 8.4 | 20.9 | 25.4 | 34.3 | **41.3** | **47.8** | 23.7 | 35.8 | **19.9** | **29.9** |
> | 0.5 | 0.8 | **_0.75_** | **25.4** | **35.2** | **22.9** | **30.3** | **25.9** | **35.4** | **23.6** | **30.5** | 10.4 | **24.1** | 8.9 | **21.4** | 26.7 | 34.8 | 40.4 | 47.4 | **24.7** | **35.9** | 19.0 | 29.5 |
> | 0.5 | 0.8 | **0.95** | 20.4 | 32.2 | 19.7 | 28.6 | 24.4 | 34.4 | 22.7 | 29.8 | **10.5** | 23.3 | **9.0** | 21.0 | **29.7** | **35.1** | 37.6 | 46.4 | 23.8 | 34.8 | 17.8 | 29.2 |

---

> ### Comment · Reviewer_UESN · 2024-11-25
>
> This reviewer appreciates the authors’ efforts on the responses and paper revision. While the authors have addressed some of the raised concerns, primarily through additional experiments and clarifications, the core issue of the proposed method's novelty and technical contribution persists. Specific comments are given as follows:
>
> Q1 and Q2: There were no changes to my initial review comments. Although the authors justified the differences between the proposed techniques, i.e., object center and boundary distance fields, and related work in the literature, this reviewer is still concerned about the high similarity between the proposed techniques and existing methods. The authors also agreed that the four-step procedure is heuristic to some degree.
>
> Q3, Q6, and Q9: This reviewer is happy that the authors took the comments into consideration when revising the paper.
>
> Q4, Q5, and Q8, and Q10: They have been resolved. This reviewer appreciates the efforts of new experiments, method modification, and paper revision.
>
> Q7: It has been resolved in part. This reviewer appreciates the efforts of new experiments to evaluate the sensitivity of hyperparameters. However, involving many hyperparameters in unsupervised learning tasks is less practical since there is no ground truth for tuning the hyperparameters.
>
> The proposed method demonstrates effectiveness, but its novelty and technical significance are limited. Thus, this reviewer keeps the rating unchanged.

---

> > ### Author Response · Authors · 2024-11-28
> > **Responses to Comments from Reviewer UESN (Part 1)**
> >
> > **Comment \#1: Q1 and Q2: There were no changes to my initial review comments. Although the authors justified the differences between the proposed techniques, i.e., object center and boundary distance fields, and related work in the literature, this reviewer is still concerned about the high similarity between the proposed techniques and existing methods.**
> >
> > **Response \#1:** The reviewer constantly mentions the similarity between our method and existing works. However, after two rounds of discussion, there is no single published work cited for us to compare. We kindly respect the reviewer's opinion. However, your criticism grounding on personal views instead of providing related publications makes it hard to address your true concern in mind.
> >
> > Regarding the novelty of our method, our team has been working on this project for more than one year, and has extensively surveyed and been constantly monitoring the literature up to date. We confirm that there is no such similar work (a three level objectness network + center-boundary aware multi-object reasoning) in the challenging field of unsupervised object segmentation to the best of our knowledge.
> >
> > Nevertheless, we are always glad to discuss the uniqueness of our method, once the reviewer is willing to share related works at hand.
> >
> > **Comment \#2: The authors also agreed that the four-step procedure is heuristic to some degree.**
> >
> > **Response \#2:** Sure. Heuristics (*i.e.*, priors or assumptions in general) have been widely used and respected in designing algorithms over decades. In the field of unsupervised object segmentation, it is always crucial to incorporate various heuristics due to the lack of human labels. For example, our baselines CutLER(CVPR'23)[2] and CuVLER(CVPR'24)[1] hold the heuristic that pixel features within an object should be similar, while the baselines TokenCut(CVPR'22)[3] and LOST(BMVC'21)[4] hold the heuristic that objects should be smaller than backgrounds.
> >
> > For our method, the four-step multi-object reasoning module holds the heuristic that an object should have its spatial center/position and be bounded by its shape boundary. Most notably, our search of object boundaries follows the mathematic properties of normalized signed distance values in Equation (6), thus being physically meaningful instead of groundless.
> >
> > Overall, we respect the reviewer's own philosophy for designing models, but incorporating heuristics into algorithms itself should not be regarded as a weakness.
> >
> > **References:**
> >
> > [1] Arica, Shahaf, et al. "CuVLER: Enhanced Unsupervised Object Discoveries through Exhaustive Self-Supervised Transformers." CVPR, 2024.
> >
> > [2] Wang, Xudong, et al. "Cut and learn for unsupervised object detection and instance segmentation."  CVPR, 2023.
> >
> > [3] Wang, Yangtao, et al. "Self-supervised transformers for unsupervised object discovery using normalized cut." CVPR, 2022.
> >
> > [4] Siméoni, Oriane, et al. "Localizing objects with self-supervised transformers and no labels." BMVC, 2021.

---

> > ### Author Response · Authors · 2024-11-28
> > **Responses to Comments from Reviewer UESN (Part 2)**
> >
> > **Comment \#3: Q3, Q6, and Q9: This reviewer is happy that the authors took the comments into consideration when revising the paper. Q4, Q5, and Q8, and Q10: They have been resolved. This reviewer appreciates the efforts of new experiments, method modification, and paper revision.**
> >
> > **Response \#3:**  Thank you for acknowledging our rebuttal materials and we are glad that your concerns have been addressed accordingly.
> >
> > **Comment \#4: Q7: It has been resolved in part. This reviewer appreciates the efforts of new experiments to evaluate the sensitivity of hyperparameters. However, involving many hyperparameters in unsupervised learning tasks is less practical since there is no ground truth for tuning the hyperparameters.**
> >
> > **Response \#4:** We agree that it is desired to have fewer hyperparameters for unsupervised tasks. In our method, there are only three key hyperparameters: $\tau^e_{conf}$ for existence, $\tau^c_{conf}$ for the center, and $\tau^b_{conf}$ for the boundary. As extensively ablated in a wide range of choices in Tables 2/3 in our first round response, we can apparently see that our method is barely sensitive to different choices of the three hyperparameters. In particular,
> >
> > - When $\tau^e_{conf}$ varies $(0, 0.95)$, the AP$^{box}_{50}$ ranges 28.1$\sim$ 32.6 on COCO* val set, and has a similar trend on the other 7 datasets.
> > - When $\tau^c_{conf}$ varies $(0, 0.95)$, the AP$^{box}_{50}$ ranges 29.8$\sim$ 32.6 on COCO* val set, and has a similar trend on the other 7 datasets.
> > - When $\tau^b_{conf}$ varies $(0, 0.95)$, the AP$^{box}_{50}$ ranges 31.2$\sim$ 32.6 on COCO* val set, and has a similar trend on the other 7 datasets.
> >
> >
> > Notably, the three hyperparameters are physically meaningful because the corresponding three-level object-centric representations are designed with explicit meanings, instead of relying on implicit hidden vectors. This is the fundamental reason that once the hyperparameters are selected, they can generalize well on different datasets without needing extra tuning.
> >
> > **Comment \#5: The proposed method demonstrates effectiveness, but its novelty and technical significance are limited. Thus, this reviewer keeps the rating unchanged.**
> >
> > **Response \#5:** Thank you for acknowledging the effectiveness of our method.
> >
> > Regarding the novelty, again, we are eagerly waiting for your reply about related publications. So far, there is no work similar to ours (three level objectness network + center-boundary aware multi-object reasoning) to the best of our knowledge.
> >
> > Regarding the technical significance, our method clearly surpasses all existing methods on 6 public datasets by large margins, achieving state-of-the-art performance, and pushing the boundaries of the challenging unsupervised object segmentation. Moreover, our solution provides a new perspective to learn and use object-centric representations. Here, \textit{learn} means the three level explicit object-centric representations, while \textit{use} means the center-boundary aware multi-object reasoning strategy. Above all, our technique contributions are new, effective, and significant for the field of unsupervised object segmentation.

---

> ### Comment · Reviewer_UESN · 2024-11-29
>
> Here are some examples of similar methods for your reference.
>
> Regarding your proposed center field, a similar technique, called the displacement field, is presented in Section 4.3 and Figure 6c of [1]. This proposed displacement field associates each pixel with a 2D vector pointing to the centroid of the object instance. It is also developed for segmentation.
>
> [1] J. Ahn et al. Weakly Supervised Learning of Instance Segmentation with Inter-pixel Relations, in CVPR, 2019.
>
> In your proposed object boundary distance field, each pixel is assigned the shortest distance to the object boundary. This technique is commonly used in distance transforms, such as Chamfer distance and its variants, as mentioned in the initial reviews. It is widely adopted for object shape and contour (boundary) representations. Some related papers [2, 3] are provided below, where distance transform and Chamfer distance are used for template matching and object detection.
>
> [2] Nguyen et al. A Novel Chamfer Template Matching Method Using Variational Mean Field, in CVPR, 2014.
>
> [3] Ma et al. Boosting Chamfer Matching by Learning Chamfer Distance Normalization, in ECCV, 2010.

---

> > ### Author Response · Authors · 2024-11-30
> > **Responses to Comments from Reviewer UESN (Part 1)**
> >
> > **Comment \#1: Regarding your proposed center field, a similar technique, called the displacement field, is presented in Section 4.3 and Figure 6c of [1]. This proposed displacement field associates each pixel with a 2D vector pointing to the centroid of the object instance. It is also developed for segmentation.\
> > [1] J. Ahn et al. Weakly Supervised Learning of Instance Segmentation with Inter-pixel Relations, in CVPR, 2019.**
> >
> > **Response \#1:** Thank you very much for sharing the related work IRNet[1]. Here we summarize the key differences between IRNet and our Object Center Field:
> >
> > - **The Definition of Displacement Field *v.s.* our Object Center Field:**
> > 	- In Section 4.3 of IRNet, in a Displacement Field, each vector of a specific pixel is defined as the displacement between this pixel and its associated object centroid, particularly including both the direction and distance between the pixel and object centroid.
> > 	- By contrast, in our Object Center Field defined in Eq. (1) of our paper, each vector of a specific pixel (within the object) is only defined as a unit length vector pointing to the object center, whereas a zero vector is defined for background pixels.
> > - **The Learning of Displacement Field *v.s.* our Object Center Field:**
> > 	- The learning for the Displacement Field, denoted as $\mathcal{D}$, relies on two constraints: (1) The distance between two vectors should be the same as their coordinate distance: $\mathcal{D}(x_1, y_1) - \mathcal{D}(x_2, y_2) = (x_1, y_1) - (x_2, y_2)$. (2) The sum of all vectors in the Displacement Field within an object should be 0: $\sum_{x,y} \mathcal{D}(x,y) = 0$.  As stated in Section 5.1 of IRNet, the learned Displacement Field often fails to predict exact offsets to centroids and thus requires further refinement.
> > 	- By contrast, our Object Center Field does not rely on similar constraints designed in IRNet, but is directly supervised by center field signals from pseudo-labels generated by self-supervised features on ImageNet.
> > - **The Use of Displacement Field *v.s.* our Object Center Field:**
> > 	-  In IRNet, the Displacement Field is used for **clustering**. Pixels are transferred to object centroids by the learned Displacement Field and then grouped based on locations.
> > 	- By contrast, our Object Center Field is used for **splitting**. As illustrated in Figure 3 of our paper, we use the learned Object Center Field to identify multi-centers within a proposal and then split it into subproposals to tackle the under-segmentation issue.
> >
> > Overall, despite sharing the term "center or centroid", our Object Center Field and the Displacement Field in IRNet are notably different in terms of definition, learning, and most importantly, the purpose of use.

---

> > ### Author Response · Authors · 2024-11-30
> > **Responses to Comments from Reviewer UESN (Part 2)**
> >
> > **Comment \#2: In your proposed object boundary distance field, each pixel is assigned the shortest distance to the object boundary. This technique is commonly used in distance transforms, such as Chamfer distance and its variants, as mentioned in the initial reviews. It is widely adopted for object shape and contour (boundary) representations. Some related papers [2, 3] are provided below, where distance transform and Chamfer distance are used for template matching and object detection.\
> > [2] Nguyen et al. A Novel Chamfer Template Matching Method Using Variational Mean Field, in CVPR, 2014.\
> > [3] Ma et al. Boosting Chamfer Matching by Learning Chamfer Distance Normalization, in ECCV, 2010.**
> >
> > **Response \#2:** Thank you very much for sharing the related works[2,3]. Here we summarize the key differences between them and our Object Boundary Distance Field:
> > - **Distance Transform in Chamfer Matching *v.s.* in our Object Boundary Distance Field:**
> > 	- As firstly proposed in [4] and improved in [2,3,5], Chamfer distance function or distance transform can be viewed as a score map for smooth shape similarity calculation. For example, given a target shape and its calculated distance transform field, a good template matched should be the one that gives a small summed score on the template pixels multiplied by the distance transform field.
> > 	- By contrast, in our Object Boundary Distance Field, the values in the field are not used as scores for shape similarity measuring, but **indicators** to guide the optimization of bounding boxes. As shown in Eq. (6) of our paper, negative values drive the bounding box to contract and positive values to expand.
> > - **Object Detection via Chamfer Matching *v.s.* our Object Boundary Distance Field:**
> > 	- As shown in [2,3], Chamfer Matching can also be used for object detection via template matching. However, this template matching can only determine to what extent a bounded contour is similar to an expected one, but it is unable to guide the optimization of bounding boxes. In this way, object detection with Chamfer Matching needs to densely sample many bounding box locations, sizes, and aspect ratios, and then try to match them with a template.
> > 	- By contrast, our Object Boundary Distance Field is used to directly optimize the object bounding box via contraction or expansion. Here, we neither have a template to match, nor need to do so.
> >
> > In summary, the distance transform in Chamfer Matching is used for shape similarity calculation, and it can be used for object detection via brute-force searching and matching. However, our Object Boundary Distance Field is utilized to explicitly optimize the bounding box, making it efficient for boundary reasoning and object segmentation.
> >
> > **Overall Response:** We appreciate the reviewer's suggestion to discuss the differences between the related works [1,2,3] and our method. We would highlight that both our Object Center Field and Object Boundary Distance Field are different in many ways such as definition, learning, and the purpose of use. Lastly, in addition to the proposed three level object-centric representations, our multi-object reasoning module is unique and effective. These modules together allow our method to achieve state-of-the-art performance in the challenging problem of unsupervised object segmentation. The related works will be discussed in our next version.
> >
> > **Reference:**
> >
> > [1] J. Ahn et al. "Weakly Supervised Learning of Instance Segmentation with Inter-pixel Relations." CVPR, 2019.
> >
> > [2] Nguyen et al. "A Novel Chamfer Template Matching Method Using Variational Mean Field." CVPR, 2014.
> >
> > [3] Ma et al. "Boosting Chamfer Matching by Learning Chamfer Distance Normalization." ECCV, 2010.
> >
> > [4] Barrow, Harry G., et al. "Parametric correspondence and chamfer matching: Two new techniques for image matching." IJCAI, 1977.
> >
> > [5] Shotton, Jamie, Andrew Blake, and Roberto Cipolla. "Multiscale categorical object recognition using contour fragments." TPAMI, 2008.

---

> > > ### Author Response · Authors · 2024-12-02
> > > **Waiting for discussion**
> > >
> > > Dear reviewer UESN,
> > >
> > > Since the discussion is closing very soon, we are still waiting for your thoughts on our latest discussion about the related works shared by you.  We would be grateful if you could share your further feedback. Thank you for your time.
> > >
> > > Best,
> > > Authors

---

### Official Review · Reviewer_7Rij · 2024-11-04

**Soundness:** 2
**Presentation:** 3
**Contribution:** 3
**Rating:** 6
**Confidence:** 3

**Summary:**

This paper proposes a two-stage pipeline consisting of an object-centric representation learning stage followed by a multi-object reasoning stage for unsupervised multi-object segmentation. The proposed three levels of objectness: 1) a binary object existence score, 2) an object center field, and 3) an object boundary distance field are used to learn object-centric representations. Given experiments show that the suggested method achieves state-of-the-art object segmentation performance.

**Strengths:**

This paper is well-presented and provides sufficient detail, making it easy to follow.

The task of unsupervised multi-object segmentation that the authors have investigated is both interesting and challenging.

Compared to the competing methods, the proposed approach demonstrates a significant performance improvement.

**Weaknesses:**

Could the authors provide more details on how the rough masks are generated from ImageNet without using human annotations? Specifically, what method is used to distinguish foreground from background, and how is this process unsupervised?

Have the authors considered how their method might be adapted for use in domains like medical or sonar imaging, where the nature of objects and backgrounds differs significantly from natural images? What modifications might be necessary to make the approach more generalizable across diverse image types?

**Questions:**

How are the rough masks obtained, and does the process require supervised training?

---

> ### Author Response · Authors · 2024-11-24
> **Responses to Reviewer 7Rij**
>
> We appreciate the reviewer's valuable comments and address the concerns below.
>
> ***Q1: Could the authors provide more details on how the rough masks are generated ... is this process unsupervised?***
>
> **A1**: We exactly follow the VoteCut method proposed in CuVLER [1] to obtain a single object mask (binary) on each image of ImageNet. First, each image of ImageNet is fed into self-supervised pretrained DINO/v2 [2,3], obtaining patch features. Second, An affinity matrix is constructed based on the similarity of patch features, followed by Normalized Cut [4] to obtain multiple object masks. Third, the most salient mask of each image is selected as the rough foreground object. For more details, refer to CuVLER.
>
> Notably, this whole process does not involve any human labels. Fundamentally, it relies on the high-quality per-pixel features obtained from DINO/v2 which is trained with successful self-supervised techniques without needing any human labels as well.
>
> In the revised paper, we clarify this process of generating rough masks in lines 157-161 of Section 3.1.
>
> ***Q2: Have the authors ... be adapted for use in domains like medical or sonar imaging, ... to make the approach more generalizable across diverse image types?***
>
> **A2**: Thanks for this suggestion. We further evaluate our method on the total 165 medical images of a gland dataset GlaS [5]. The following Table 1 shows the quantitative results and Figure 20 in the Appendix of the revised paper shows qualitative results. We can see that our method achieves higher scores than baselines in both settings of direct object discovery and zero-shot detection.
>
> Nevertheless, the scores of all methods are clearly lower than other datasets of natural images, essentially due to the significant domain gaps between natural images and medical images. We leave this non-trivial problem for future exploration.
>
> In the revised paper, we add our quantitative results on GlaS dataset in Table 3 of Section 4.3, and qualitative results in Appendix A.15.
>
> *Table 1: Gland segmentation results for MaskCut/CutLER, VoteCut/CuVLER, and OCN $_{\text {disc }}$/OCN, under direct object discovery and zero-shot detector setting.*
> |  |  | $\mathrm{AP}_{50}^{\text {mask }}$ | $\mathrm{AP}_{75}^{\text {mask }}$ | $\mathrm{AP}^{\text {mask }}$ | $\mathrm{AR}_{100}^{\text {mask }}$ |
> | :---: | :---: | :---: | :---: | :---: | :---: |
> | MaskCut $(\mathrm{K}=3)$ | direct object discovery | 0.4 | 0.1 | 0.2 | 0.8 |
> | MaskCut $(\mathrm{K}=10)$ | direct object discovery | 0.4 | 0.1 | 0.2 | 0.9 |
> | CutLER | zero-shot detector | 8.8 | 1.0 | 2.6 | 21.5 |
> | VoteCut | direct object discovery | 0.8 | 0.0 | 0.2 | 1.9 |
> | CuVLER | zero-shot detector | 3.2 | 0.2 | 0.7 | 11.1 |
> | OCN $_{\text {disc }}$ (Ours) | direct object discovery | 3.3 | 1.6 | 1.7 | 6.8 |
> | OCN (Ours) | zero-shot detector | 9.6 | 1.2 | 2.9 | 18.9 |
>
> ***Q3: How are the rough masks obtained, and does the process require supervised training?***
>
> **A3**: Refer to A1.
>
>
> **References:**
>
> $[1]$ Shahaf Arica et al., CuVLER: Enhanced Unsupervised Object Discoveries through Exhaustive Self-Supervised Transformers, CVPR 2024
>
> $[2]$ Caron, Mathilde, et al. "Emerging properties in self-supervised vision transformers.". ICCV 2021.
>
> $[3]$ Oquab, Maxime, et al. "Dinov2: Learning robust visual features without supervision." Transactions on Machine Learning Research, 01/2024.
>
> $[4]$ Shi, Jianbo, and Jitendra Malik. "Normalized cuts and image segmentation." IEEE Transactions on pattern analysis and machine intelligence 22.8 (2000): 888-905.
>
> $[5]$ Sirinukunwattana, Korsuk, et al. "Gland segmentation in colon histology images: The glas challenge contest." Medical image analysis 35 (2017): 489-502.

---

> > ### Comment · Reviewer_7Rij · 2024-11-25
> >
> > Thanks for the further analyses of rough masks and the experiments on medical image.
> >
> > Like the author, it is noted that the scores are clearly lower than natural images. But the suggested method is unsupervised, so the main reason for the low score cannot be attributed to the significant domain gap between natural images and medical images.  If the proposed method still requires domain specific data for pre training, this will weaken the advantage of such the unsupervised method.
> >
> > Overall, the authors have addressed most of my concerns, and  I will maintain my initial positive rating.

---

> > > ### Author Response · Authors · 2024-11-26
> > > **Thanks**
> > >
> > > Dear reviewer 7Rij,
> > >
> > > Thank you very much for following up on our rebuttal materials and keeping your valuable positive rating.
> > >
> > > Yes, we agree that the scores of all existing unsupervised methods are rather low. This highlights that your suggested task (unsuperivsed training + zeroshot test) is extremely chanllenging. In this hard setting, our method still achieves the highest score on AP50. We hope that our work could inspire more advanced methods to tackle the hard problem of unsupervised learning.
> > >
> > > Best,
> > > Authors

---

### Author Response · Authors · 2024-11-25
**Overall Responses**

We appreciate insightful comments and valuable suggestions from all reviewers. A revised paper with an appendix is presented. Specifically, we highlight the revised content in yellow. Revisions include:

**Presentations:**

 - Corrected typos.

 - Integrated illustration figures for three-level representations (Figure 1).

 - Updated center reasoning description with connected-component processing (lines 283 - 289).

 - More distinguishable names for our methods under different protocols: OCN$_{disc}$ for protocol 1, OCN for protocols 2 and 3 (lines 353 - 356).

 - More detailed qualitative results (Figure 5, Figure 14-18).

 - Elaborations for experiment settings (lines 364-375).



**Experiments:**

 - Additional evaluation metrics for more insight into experimental results (Table 1).

 - Experiments on medical images (Table 3, Appendix A.15).

 - Ablation studies on a wider range of hyper-parameters on 7 datasets (Table 10, Table 11).

 - Ablation studies on the effectiveness of random-cropping augmentation (Table 12).



**Discussions:**

 - Clarifications for the novelty and design principles of our method (lines 182 -186, lines 97-98).

 - Discussions on the superiority of our method compared to existing works (lines 139-145).

 - Details on rough mask generation (lines 157 -161).

 - An in-depth analysis of the experimental results (lines 391-402, Table 1).

 - More information on the efficiency of proposal optimization (Appendix A14).

---

### Meta-Review · Area_Chair_wRFg · 2024-12-16

**Metareview:**

This paper receives 1 negative rating and 3 positive ratings (while one positive review is very short). Although the paper has some merits like competitive results, the reviewers pointed out a few critical concerns about 1) limited novelty and technical contributions, 2) cross-domain gaps, 3) deeper analysis of experimental results. After taking a close look at the paper, rebuttal, and discussions, the AC agrees with reviewers' feedback and hence suggests the rejection decision. The authors are encouraged to improve the paper based on the feedback for the next venue.

**Additional Comments On Reviewer Discussion:**

In the rebuttal, some of the concerns like technical clarity are addressed by the authors. However, during the post-rebuttal discussion period, the reviewer UESN is not convinced about the technical contributions of the proposed method, especially on related work of using objectness (e.g., object center and boundary distance fields), in which the similar concerns are also pointed out by reviewer gBo7. In addition, the reviewer 7Rij mentioned the cross-domain gap issue in the review, while the authors have not addressed it well in the rebuttal. Although the reviewer did not argue it much, the AC thinks this is still a critical issue to be answered (e.g., whether the method requires domain-specific data for training to obtain more reasonable results). Considering that the reviewer ffrn only provided a much shorter review, the AC took a close look at all the other contents and agrees that the authors have not addressed the concerns of technical comparisons with the prior work well, in which the paper still requires to be significantly improved before making it ready for publication.

---

### Decision · Program_Chairs · 2025-01-22

Reject

---

> ### Public Comment · ~Bo_Yang7 · 2025-07-02
> **Accepted by ICML 2025**
>
> The updated version has been published at ICML 2025. Welcome to check out our paper and code.
>
> Full paper: https://arxiv.org/abs/2506.01778
>
> Code and data: https://github.com/vLAR-group/unMORE